# MILCA: Multiple Instance Learning using Counting and Attention

## Abstract

In Multiple Instance Learning (MIL), a bag is comprised of instances and the label is prescribed to the whole bag, with no information on the labels of each instance. The leading approaches for MIL are Embedded Space (ES) solutions, where the full bag is embedded into a vector space. While very complex models were constructed for MIL classification tasks, we show that often some features are associated with a class, and a simple counting/summing algorithm leads to similar or better accuracy than current solutions. This can be improved in some cases by weighting these selected features using a fully connected network to predict the coefficient of each feature. However, a simple relative contribution of each feature, where the sum of the coefficients is normalized to 1, fails to count the feature. Thus instead, we replace the softmax by a projection of the coefficients to [-1,1] or [0,1] but do not limit their sum. This allows the model to count features. The resulting algorithm - MILCA (Multiple Instance Learning using Counting and Attention) is applied to multiple previous and new real-world MIL tasks, as well as recovering the host disease history from sequenced T Cell Receptor Repertoires. In most cases, MILCA is significantly better and way more efficient than currently used MIL algorithms, with a 3 % higher accuracy than current SOTA on average. To summarize, in MIL classification tasks, where often the number of features is large compared to the number of bags, complex models are typically not better than a weighted sum of informative features. The code for MILCA is available at: github.com/submissionanonymous6/MILCA

## 1 Introduction

Multiple Instance Learning (MIL) is the classification of a bag composed of instances, where no information on the class of each instance is available (Dietterich et al., 1997). While the original focus of MIL has been on Tabular (Dietterich et al., 1997), text (Zhou et al., 2005), and image classification (Andrews et al., 2002), recently more problems are treated as MIL, including sound event detection, tumors recognition (Wang et al., 2023b; Sudharshan et al., 2019), and even deep fake video detection (Li et al., 2020).

In this context, one can define the MIL classification problem as follows: A bag $B = \{\bar{x}_1, \cdots, \bar{x}_n\}$ consists of instances $\bar{x}_i$. Each instance is represented by a feature vector $(x_i^1, \cdots, x_i^m)$ (that may have originated from an image or any other data). The bag has a label $y_B$. This label is often binary but may be categorical or continuous (Romeo et al., 2019). The bag's instances may also have unseen labels, $y_{x_i}$ that are unavailable during the training phase.

Classically, in strict MIL problems, a bag $B$ has a positive label if and only if at least one of its instances has a positive label (Dietterich et al., 1997), i.e. $:y_B = 1 \iff \exists \bar{x}_i \in B : y_{\bar{x}_i} = 1$. However, often, a more lenient definition is used, where we only assume that the distribution of instances differs between the positive and negative bags(Foulds & Frank, 2010). We here use this more lenient definition. Three main types of solutions were proposed for MIL problems: Instance Space (IS), Bag Space (BS), and Embedded Space (ES) solutions (Amores, 2013).

IS solutions are based on the assumption that bag labels are connected to their instance labels (Dietterich et al., 1997; Andrews et al., 2002). Therefore, one first estimates the labels of the instances for each bag, and only then evaluates the label of the bag itself using a classifier $F(\hat{y_{x_1}}, \cdots, \hat{y_{x_n}})$, where $\hat{y_{x_i}}$ is the estimated class of the instance. For example, a book (a bag) can be labeled as good

or bad. The book has multiple chapters (instances) and each chapter has words (features). In IS solutions, one first labels the book chapters as good or bad, and only then evaluates the book label.

BS Solutions treat each bag as an entity. The classification is performed using the similarity between different bags using various distance functions on the bags $D(B_i, B_j)$ (Wang & Zucker, 2000). In BS, one generally cannot determine if an instance is associated with a class. For example, one can compare books by their length, word count, mean word length, or other parameter.

Finally, ES solutions rely on the projection of the entire bag into a vector space (in contrast with the projection of each instance) before performing a classification (Wei et al., 2016). In ES, even if the projection is at the instance level, the instance projections are combined into a bag projection(Ilse et al., 2018; Tarkhan et al., 2022). In the book classification problem above, the book is projected into a set of parameters of the book, either predefined or learned. ES solutions are argued to be more accurate than IS (Wang et al., 2018).

## 2 NOVELTY

The main limitation of conventional ES solutions lies in their inability to effectively integrate feature-level and bag-level information. In most MIL datasets, the feature dimension is high (e.g., words in a bag of words or pixels in an image), only a small subset of features is relevant to the class, and the number of bags is limited. Consequently, current ES classifiers are prone to overfitting, and simpler models outperform more complex ones.

In this work, we argue the following:

1. Overfitting can be mitigated through straightforward feature selection by identifying features associated with one of the classes (there is typically an imbalance, where features are positively associated with only one of the classes, as will be further discussed). Counting or averaging the relevant features across instances creates a simple classifier.

2. This approach leads to a classifier we denote as MILCA (Multiple Instance Learning using Counting and Attention). MILCA is both significantly faster and often more accurate than State-Of-The-Art (SOTA) models on a variety of real-world datasets, including a set of novel datasets derived from Wiki pages. With virtually no learnable parameters, MILCA also offers a highly interpretable model structure. The enhanced accuracy of MILCA demonstrates that, in many cases, complex models are unnecessary.

3. One can combine MILCA with a simple feed-forward network (FFN). The last layer of the FFN can be improved by limiting each weight to be in the range of [0,1] or [-1,1] (depending on the importance of negative weight), instead of setting their sum to be 1. In practice, we replace a softmax with a sum of TANH on each feature: $y = \sigma(\sum tanh(\alpha_i) * v_i)$, where $\alpha_i$ are learned weights, and $v_i$ are the features. This change can allow the layer to count events.

We also propose a new set of MIL classification tasks, based on the detection of keywords associated with Wiki pages, and each instance is the bag of words of a section of the wikipage.

## 3 RELATED WORK

We here focus on ES solutions. Reviews are available for IS and BS solutions (Dietterich et al., 1997; Andrews et al., 2002; Wang & Zucker, 2000). The leading models for MIL classification (that are ES solutions) use tailored algorithms. miVLAD and miFV (Wei et al., 2016) are using the VLAD (Jégou et al., 2010) and FV (Sánchez et al., 2013) computer-vision based encoders. These encoders turn each bag $B$ into a high-dimension vector $(b_1, \cdots, b_n)$. VLAD clusters the instances using K-means and represents each instance as its distance from its centroid. FV represents each bag as its Fisher Vector representation. Both models classify the bag using an SVM.

Attention-based ES models, such as Instance Attention and Gated Attention (Ilse et al., 2018; Tarkhan et al., 2022) have a high accuracy in multiple classification tasks. Both Instance Attention and Gated Attention are based on the concept that some instances have less information on the bag labels. Both models use an attention layer as the encoder, creating a single vector $\bar{b}_E$ that

represents its corresponding bag. The models then classify the vectors using a unique MIL pooling technique (Ilse et al., 2018), and a fully connected neural network (Tarkhan et al., 2022). Recently, SMILES (Wang et al., 2023a) claim to outperform all recent models using a useful representation of the bags through self-supervised learning. SMILES generates a multi-instance graph for each bag and encodes the bag's graph using message-passing layers and an ordered weighted averaging operator. SMILES does not offer a code or details to allow for a comparison of their results.

The concept of 'counting' has been previously discussed in MIL problems. However, it represents different ideas in different applications. For example, Hajimirsadeghi et al(Hajimirsadeghi et al., 2013) focus on counting the labels of instances within a bag, emphasizing the proportion of positive versus negative instances to guide classification, in contrast with MILCA that emphasizes counting to detect informative features (not instances) across instances. Hajimirsadeghi et al(Hajimirsadeghi et al., 2015) count instance labels to enrich kernel functions to define similarities between bags. This process aligns more with traditional statistical approaches that abstractly handle data relationships and can also be considered a Bag Space solution. The work most similar to MILCA is Raykar et al (Raykar et al., 2008) which also performs feature selection. However, it uses this to produce a prior on a bag of words classification. Note that a similar terminology, yet a different approach was used in the context of instance classification(Foulds & Frank, 2010). However, it is a different approach, since MILCA does not classify instances, but counts to choose features.

In contrast, MILCA distinctively counts and sums important feature occurrences among the different instances directly, using these counts to construct a straightforward and interpretable vector representation for each bag.

## 4 MODEL

We present three MILCA variants with three modular stages (Figure 1). The variant only differ in the last stage.

A) Translation of each instance to a vectorial representation (Fig. 1A) if the instances are not already in that form. Note that in many of the real-world datasets studied, this stage was already performed as a pre-processing step (in published datasets). The embedding was then taken from the original publication for each dataset.

B) A subset of the features associated with a class is selected (Fig. 1B). Features are associated with a class if they are higher on average (or more frequent) in instances in bags of a class. Formally, for each feature $k$ in bag $B$ $z_B^k = average_{i \in B}(x_i^k)$. A Mann-Whitney test is performed between $z_B^k$ in instances belonging to each class (or Fisher exact test in the case of counting). Features with a p-value below a threshold, or simply the top $M$ most significant features ($p$ and $M$ are optimized per dataset on the training set) are selected (further denoted as $SF$). Each significant feature is associated with the class where its average is highest (further defined as positive features $PF$ and negative features $NF$).

C) Counting. Once the informative features are selected, MILCA counts (in the discrete case) or sums (in the continuous case) the average in each bag of instances "associated" with each class (Fig. 1C). After that, we employ a min-max normalization to standardize the feature scales across the training bags, ensuring that no single feature disproportionately influences the model's output. We propose three possible classifiers to classify the bag embeddings:

Given a set of bag representations $z_B^k$, C1 is a naive counter that sums only features associated with the class that has the largest $PF$ group:

$$C1(B) = \sigma(\sum_{k \in PF} z_B^k) \tag{1}$$

In this classifier, no learning or optimization is performed, and the training set is only used to detect the informative features. In C2, we use the weighted difference between the sums of features associated with each class

$$C2(B) = \sigma(\sum_{k \in PF} z_B^k) - \beta\sigma(\sum_{k \in NF} z_B^k). \tag{2}$$

In this classifier, the training set is used to detect the informative features and to optimize $\beta$ to maximize the Area Under Curve (AUC) of the training set. For the $C3$ classifier, we train a two-

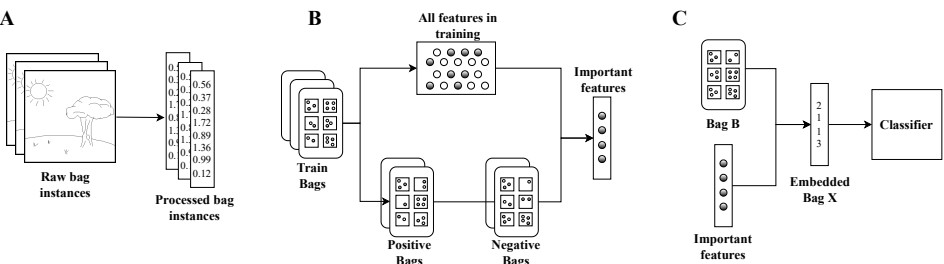

Figure 1: **A.** Translation of each instance into a vectorial representation. This part is not trained but is a part of the preprocessing. We do not limit the dimension. In the TCR example below, the dimension is over $10^8$. **B.** Selection of features associated with each class, using a comparison of the feature average between bags of each class. **C.** Using a (weighted) sum to classify each Bag

layer FCN for the coefficient of each feature, and then classify using a product of the weights by the feature. However, simply applying a softmax on the coefficients as in typical attention mechanisms fails to count the positive and negative features. Thus, instead, where each weight is transferred to the [-1,1] or [0,1] range using a tanh or a sigmoid:

$$C3_\sigma(B) = \sigma(\sum_k \sigma(\alpha_k)z_B^k); \; C3_{tanh}(B) = \sigma(\sum_k tanh(\alpha_k)z_B^k) \qquad (3)$$

This is the only component that requires training. Note that in $C1$, $\sigma(\alpha)$ is simply a vector of ones, and in $C2$ it is 1 for the first class and $-\beta$ for the second class. This is equivalent to very high $\alpha_k \approx \inf$ for the first class and very negative for the second class.

## 5 EXPERIMENTAL SETUP

All the presented results are the average over 10 training/validation/test divisions. Each division led to different informative features. In each division, We randomly divided each data set into training (64%), validation (16%), and testing (20%), except for the $C1$ counter, where the validation and training were merged. The significant features were determined using the training and validation sets. We trained the $C3$ model using an Adam optimization and a binary cross-entropy loss function and added two fully connected layers before the final classifier. The hyper-parameters were optimized for each division and each dataset by itself, to reach the maximum AUC score on the internal validation set. Following parameter optimization, the models were trained on the combined training and validation sets and tested on the test set. Lastly, we evaluated the models performance on the test set using Area Under ROC Curve (AUC) and accuracy. While the accuracy metric is not optimal for imbalanced datasets, we report it here to ensure consistency with previous publications.

The following algorithms were compared: mi-SVM and MI-SVM (Andrews et al., 2002), EM-DD (Zhang & Goldman, 2001), MIVLAD and mi-FV(Wei et al., 2016), MI-Kernel (Gärtner et al., 2002), mi-Graph (Zhou et al., 2009), mi-Net and MI-Net (Wang et al., 2018), Attention and Gated Attention (Ilse et al., 2018), BGraph (Pal et al., 2022). SMILES (Wang et al., 2023a), BDR (Huang et al., 2022) , and MI-SDB (Feng et al., 2021) were not used as we could not validate their results, since no functioning github was available.

We compared MILCA, which was implemented using the PyTorch python package, to all of the publicly available models that have available functioning githubs. The list of models is detailed in Table 2 and Table 3 and in the github. All the models' parameters are detailed in the github repository.

## 6 RESULTS

### 6.1 INFORMATIVE FEATURES

We first argue using a simplified model that some features have an average difference between the classes that can be easily detected by a two-class comparison (t-test or Mann Whitney or Fisher exact test - here we used a t-test). We show that for a large enough number of bags ($N_B$), the fraction of False Positive ($FP$) can be arbitrarily small. Similarly, even for a small number of bags, for a strong enough association, the False Negative ($FN$) fraction is small. Combining these two arguments leads to a straightforward feature selection and counting argument that can reach any required level of sensitivity and specificity for a high enough dimension and number of bags. Surprisingly, these conditions hold for most real-world datasets classically used in MIL classification. We analyzed a simple model where all features are Bernoulli variables with an average of $\delta$, except for a subset of the features (chosen randomly with a probability $\eta$) with a different average in the positive and negative bags ($\delta_2 \in [\delta \pm \gamma]$ vs. $\delta$). We denote these features as informative features ($IF$). We simulated $N_B$ bags (See methods). The number of type I errors (False Positives $FP(N_B, \delta_1)$) when using a T-test with a cutoff of $T$ on the average of each feature in each bag is equivalent to simply treating each instance as independent, as such:

$$FP(N_F, N_B) \approx N_F(1 - \eta)ERF(\sqrt{\delta * N_B}). \quad (4)$$

The missed informative features obey $|\delta - \delta_2| < c(N_B)$ for some $c(N_B)$, and the probability of $FP$ is inversely proportional to $\gamma$. The expected number of informative features detected ($TP(N_F, \eta, \gamma)$ is the product of the total Informative features and the probability of not missing them:

$$TP(N_F, \gamma) \approx \eta N_F(1 - c(N_B)/\gamma) \quad (5)$$

Thus, one can get any required ratio between TP and FP for high enough $\gamma$ and $N_B$ (Fig. 2 left plots). In a more complex distribution, the T-test can be replaced by non-parametric or multi-class tests. These results are not sensitive to the instance number distribution details or the feature distribution, as long as the variance is limited.

To show that this argument holds in real-world data, we studied, multiple standard MIL datasets originating from either images or molecules (Dietterich et al., 1997; Andrews et al., 2002). These datasets have been extensively tested using a vast array of MIL algorithms. The same claims are supported in the real datasets, where we compared the informative features on the training set and estimated them on a test set (Fig. 2 right plot).

One can associate each candidate IF with a class, based on the difference between the average of the feature in each class (or the median in skewed distributions), producing a set of features associated with each class (Positive and Negative Features - $PF$ and $NF$ for a binary case) (Fig. 2 right plot). Interestingly, in real datasets, often, there was only one class with features associated with it.

### 6.2 COUNTING IS ALMOST ALL YOU NEED

Assuming independence over the features, the sum (count) over FP - $C(B) = \sum_{k \in PF} z_B^k$ converges to the expected sum, and thus, if $[E(\sum_{k \in PF} Z_k^B) - E(\sum_{k \in NF} z_k^B)]/\sqrt{(\sum_{k \in PF \cup NF} \sigma^2(z_k^B)}$ is large enough, $C$ can reach any required accuracy. We here show that surprisingly, there is practically no regime where this simple classifier is not accurate enough, and more complex classifiers are accurate.

We thus propose to simply sum the average over instances of $PF$ (or count them in the discrete case). This can be done at two initial levels: by taking only the most informative classes and simply summing the instances associated with this class (Later denoted as $C1$), or by using the weighted difference between the sum of instances associated with each class, with a single parameter $\beta$ - the weight of the second class (Later denoted as $C2$).

We test these methods on a simulated dataset, allowing us to control the various attributes described above. We divided the dataset into training and test sets. For $C1$, the training set was used only to extract the significant features. For $C2$, an internal validation set was used to determine the values of $\beta$ producing the highest Area Under Curve (AUC). We then applied the two algorithms to the

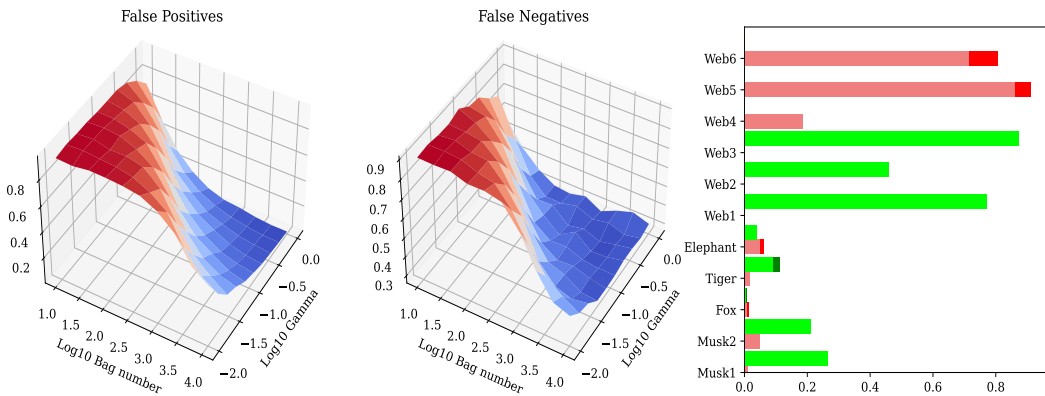

Figure 2: Left plots - False Positives and False Negatives in the simulated models as a function of the number of bags ($N_F$) and the difference between positive and negative bags in informative features ($\gamma$). For high $N_F$ or $\gamma$ values, both types of errors decrease to 0. Right plot - Feature associated in real-world datasets with positive (Green) or negative (Red) class. The differences between deep and light red/green are the non-detected informative features. This fraction is very small.

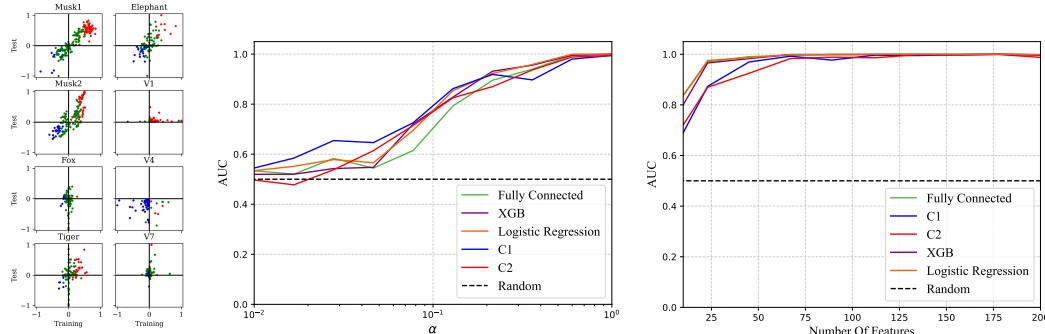

Figure 3: Left plot. Each subplot is a different dataset, and each point is a feature. The x/y axes are the average difference between classes in the training/test sets. Green points are not significant in the training. Blue/Red points are associated with the positive/negative class. Blue/Red points in the first or third quarter are consistent in the training and test sets. Right plots. Performance comparison on simulated data in benchmark models. AUC on the Y-axis vs $N_F$ (Right plot) or $\gamma$ (Left plot). We compared standard machine learning models XGboost, Logistic regression, or Fully Connected neural networks with the $C1$ and $C2$ models. The simple $C1$ and $C2$ counting are typically better or equal to all others.

simulated set above, with different values of $\gamma, N_F, N_B, \gamma$ and $\eta$. With no loss of generality in $C1$, we counted the positive class (Fig, 4). We compared the results to a fully connected network, an XGBoost, and a logistic regression. More details regrading the simulated data can be found in the datasets section. One can see that above a certain number of significant features ($N_F\eta$), the classification accuracy is perfect, and at a low number of IF, the AUC is random. The range between the two is minimal. As such, the range where complex models could help, and counting would fail is limited. If there are no informative features, the AUC is very low, and as soon as a few informative features can be found it is perfect. We further compared $C1$ and $C2$ on real-world datasets to a vast array of algorithms (See methods). Most current algorithms report the accuracy of the test. We thus report the same values for MILCA. The $C1$ algorithm is typically worse than the SOTA, but better than most reported algorithms. The $C2$ algorithm is typically 2 percent more accurate than $C1$ and often, it is better than the majority of algorithms (Table 2).

### 6.3 SOME ATTENTION MAY HELP

The $C1$ and $C2$ models treat all features equally. However, some features may be more informative than others. A simple solution to that would be a logistic regression. However, surprisingly, logistic regression has a lower accuracy than $C2$ and often of $C1$, even when a ridge component is added. Similarly, Fully Connected networks have a lower accuracy than the current models tested (Fig. 3).

We thus propose an attention-like model that produces a weighted average (Ilse et al., 2018; Tarkhan et al., 2022). However, a weighted average is not sensitive to the number of elements that you sum, only to their relative contribution, which may fail when the number of features is important. To address that, we also propose a generalization of $C1$ and $C2$ that adds a weighting function between $-1$ and $1$ to each feature independently. This is in contrast for example with a ridge loss, where the sum of the weights is regularized, but not each weight by itself, or a logistic term that transfers the output range to $[0, 1]$, with no limit on each weight. We tested such a combination by first applying the feature selection. On these features, we computed weights using a two-layer FCN. These weights were then combined with the features using Eq. 3. Indeed the results are better or equal to the current SOTA for all datasets ($C3$ in Table 2), and have 4 % more accuracy on average than the current best model, with much fewer parameters and a much shorter runtime (Table 2).

### 6.4 EXISTING AND NEW REAL WORLD AND SIMULATED DATA

The MUSK1 and MUSK2 datasets (Dietterich et al., 1997) represent molecules identified as musks or non-musks by human experts. There are 166 features in each instance that represent the shape, or conformation, of the molecule.

The Fox, Tiger, and Elephant datasets (Andrews et al., 2002) are subsets of the Corel image retrieval dataset, separated into images containing (or not) the corresponding animal. Each image was divided into segments, grouped into bags, and represented as a 230-d feature vector of the segment's color, texture, and shape. If any image segments contain the animal, the bag is labeled positive.

The Web datasets are based on web index recommendations generated for research purposes (Zhou et al., 2005). The datasets were generated mainly on web index pages from the "Yahoo", "CNN", and "FoxNews" websites. Nine volunteers categorized 113 web index pages based on their interests, resulting in nine distinct datasets. A web index page is positive if at least one volunteer expressed interest in any of its linked pages. Each web index page has 4 to 200 linked pages, grouped into bags, and described by the 1st to 20th most frequent terms appearing in the corresponding linked page, with the terms' frequencies.

Given the limited size and number of existing MIL datasets, we introduced here a novel dataset developed for this analysis. The Wiki dataset is structured as a traditional MIL, where each bag corresponds to the contents of a Wikipedia page representing a city, with a particular focus on pages about European countries. In this dataset, each bag is formed by the different sections of a Wikipedia article, treating each section as an instance within the bag. The features are a bag of words (BOW), combining all the words from all the bags except for the non-relevant words (stop words, etc.). Then, we counted all the words in the BOW for each instance. Each bag is assigned a binary label of its country. The dataset and the appropriate scrapper are available at this paper's GitHub.

### 6.5 DISEASE CLASSIFICATION USING T CELLS REPERTOIRES

An interesting application of the current counting method is the prediction of diseases from immune cell repertoires. The immune memory repertoire encodes the history of present and past infections and immunological attributes of the individual. As such, multiple methods were proposed to use T-cell receptor (TCR) repertoires to detect disease history(Emerson et al., 2017; Liu et al., 2022; Katayama & Kobayashi, 2021; Akerman et al., 2023). Each repertoire contains $10^5 - 10^7$ different TCRs, and the overlap between any two repertoires is less than $1\%$. Each TCR can be treated as a feature. Some TCRs are associated with target pathogens (CMV in the current case). Although recent models were developed to predict TCR-Antigen binding(Springer et al., 2021; 2020), those are not accurate enough to detect in advance such TCRs. Thus, there is currently no appropriate method to determine biologically informative TCRs. Additionally, there is no known mechanism for a negative association between a TCR and a disease. As such, the only applicable models are $C1$ and $C3$.

Table 1: A comparison of the performance of recent models on benchmark datasets. The given values are the accuracy percentage. The highest accuracy on each benchmark is highlighted in bold. The thick line separates the historical models, and the models presented here. All values are mean +/- 2 standard errors.

| Model | MUSK1-2 | FOX | TIGER | ELEPHANT | WEB 1-4 | Avg ↓ |
|---|---|---|---|---|---|---|
| EM-DD | 85.9 ± 4.6 | 60.9 ± 4.5 | 73.0 ± 4.3 | 77.1 ± 4.3 | N/A | 76.56 |
| MI-SVM | 81.1 ± N/A | 57.8 ± N/A | 84.0 ± N/A | 84.3 ± N/A | N/A | 77.66 |
| mi-SVM | 85.5 ± N/A | 58.2 ± N/A | 78.4 ± N/A | 82.2 ± N/A | N/A | 77.96 |
| MI-VLAD | 87.15 ± 4.3 | 62.0 ± 4.4 | 81.1 ± 3.9 | 85.0 ± 3.6 | 82.7 ± 7.3 | 80.85 |
| mi-Net | 87.35 ± 4.4 | 61.3 ± 3.5 | 82.4 ± 3.4 | 85.8 ± 3.7 | 81.6 ± 7.0 | 80.97 |
| MI-Kernel | 88.65 ± 2.3 | 60.3 ± 2.8 | 84.2 ± 1.0 | 84.3 ± 1.6 | N/A | 81.22 |
| Gated-Attention | 88.15 ± 4.6 | 60.3 ± 2.9 | 84.5 ± 1.8 | 85.7 ± 2.7 | N/A | 81.36 |
| MI-Net | 87.3 ± 4.4 | 62.2 ± 3.8 | 83.0 ± 3.2 | 86.2 ± 3.4 | 82.4 ± 9.1 | 81.40 |
| Attention | 87.5 ± 4.4 | 61.5 ± 4.3 | 83.9 ± 2.2 | 86.8 ± 2.2 | N/A | 81.44 |
| MI-Net (RC) | 88.55 ± 4.4 | 61.9 ± 4.7 | 83.6 ± 3.7 | 85.7 ± 4.0 | N/A | 81.66 |
| mi-FV | 89.65 ± 4.2 | 62.1 ± 4.9 | 81.3 ± 3.7 | 85.2 ± 3.6 | 83.1 ± 4.7 | 81.83 |
| MI-Net (DS) | 88.4 ± 4.3 | 63.0 ± 3.7 | 84.5 ± 3.9 | 87.2 ± 3.2 | N/A | 82.30 |
| mi-Graph | 89.6 ± 3.6 | 62.0 ± 4.4 | 86.0 ± 3.7 | 86.9 ± 3.5 | N/A | 82.82 |
| MILCA C1 | 81.35 ± 5.9 | 55.2 ± 7.8 | 79.7 ± 6.2 | 82.2 ± 3.0 | 93.4 ± 10.4 | 78.92 |
| MILCA C2 | 82.95 ± 5.9 | 59.0 ± 6.8 | 81.8 ± 5.4 | 83.5 ± 3.4 | 93.7 ± 11.6 | 80.65 |
| MILCA C3 (tanh) | 87.35 ± 5.5 | • **67.2 ± 3.6** | 87.5 ± 2.4 | 88.5 ± 3.0 | 96.5 ± 0.8 | 85.7 |
| MILCA C3 ($\sigma$) | • **90.45 ± 4.8** | 66.0 ± 4.0 | • **87.8 ± 1.8** | • **89.0 ± 3.6** | • **98.0 ± 2.4** | • **85.9** |

Table 2: Performance metrics of publicly available models on the Wiki dataset, the last column presents the number of Learnable Parameters each model has. The run time is on a standard i9 CPU.

| Model | Wiki | | | |
|---|---|---|---|---|
| | ACC | AUC | RT (s) | #LP |
| MI-SVM | 78.7 ± 2.4 | 94.5 ± 1.8 | 0.8 ± 0.12 | 1,783,370 |
| mi-SVM | 76.2 ± 3.2 | 91.4 ± 1.6 | 9.11 ± 1.68 | 2,541,623 |
| Attention | 90.0 ± 2.0 | 90.8 ± 3.8 | 14.013 ± 0.82 | 706,258 |
| Gated-Attention | 82.5 ± 2.6 | 87.8 ± 0.4 | 16.266 ± 1.72 | 770,386 |
| MILCA1 | 77.8 ± 1.9 | 57.6 ± 1.9 | • **0.55 ± 0.001** | • **0** |
| MILCA2 | • **91.0 ± 1.0** | • **94.6 ± 1.0** | 0.599 ± 0.003 | 2 |
| MILCA3 ($\sigma$) | 86.0 ± 3.3 | 71.8 ± 7.8 | 0.722 ± 0.184 | 4,370 |
| MILCA3 (tanh) | 90.0 ± 3.1 | 83.6 ± 6.5 | 0.678 ± 0.162 | 4,370 |

The Emerson TCR dataset contains 786 immune repertoires (Emerson et al., 2017). An immune repertoire is the collection of the T-cell receptors in a patient's blood sample. Each repertoire is labeled whether the patient was infected with CMV. Each TCR can be treated as a feature to predict the repertoire's CMV status. In this dataset, the embedding takes into account the composition of each TCR as previously described (Akerman et al., 2023).

We analyzed the immune repertoire of CMV-positive and negative patients(Emerson et al., 2017). We split the data into a training:validation:test split ratio of 8:1:1, and used 9 cross-validations on the training and validation. We then applied the $C1$ and $C3$ models. The feature selection was performed using a $\chi^2$ score for each TCR in the training set since there is a single instance for each bag and the instances are one-hot values. For the $C3$ classifier, the scoring of each TCR was expanded to a more complex network that incorporates an embedding of each TCR to incorporate the information available in the TCR sequence(Akerman et al., 2023). The $C1$ outperformed all published models, including the (Emerson et al., 2017) model on the same test set for different training set sizes (Fig. 4). The advantage of the counting algorithm is further obvious in small training sample sizes. In contrast with Emerson (Emerson et al., 2017) and deepRC (Widrich et al., 2020), the counting method can obtain a signal even for 100 training samples.

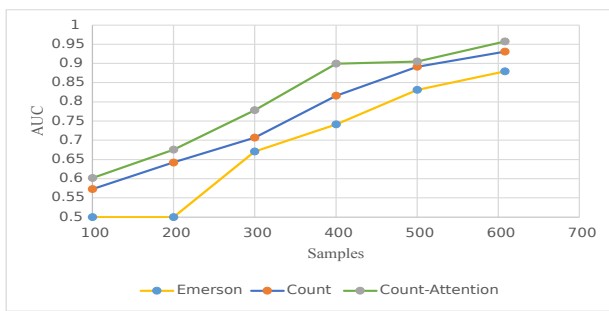

Figure 4: AUC values for different training set sizes for the detection of CMV using TCRs of $C1$ and $C3$ models compared with the current SOTA (Emerson).

## 6.6 Simplified simulated data

We simulated a simplified dataset. The generator produces two sets of $N_B$ bags each labeled positive or negative. Each bag was composed of instances with $N_F$ features. The number of instances per bag was a $\lambda = 3$ Poisson random variable. Empty bags were replaced by a single instance. Each feature in the positive and negative bags was a Bernoulli variable with an average of $\delta$, except for a subset of features in the positive bag with a random average value distributed uniformly in the range $\delta \pm \gamma$. The features in this subset were chosen independently with a probability of $\eta$.

## 7 Discussion

**Summary.** Very complex methods were proposed to integrate advanced ML approaches in MIL problems. We have here shown that a simplistic approach is better than current models for most datasets. The basic explanation is that MIL problems are typically rich in data, but with a limited number of samples. As such, almost any learning performed is an over-fit. Moreover, there is a very limited range of parameters where the simple counting arguments fail but other more complex algorithms may succeed. Thus, for many realistic scenarios, either the counting works and more complex models are an overfit, or there is no information and no algorithm works. This is not to say that a complex adapted model cannot perform better than simple counting or weighted counting, but that the difference is typically not large.

Beyond the counting we propose an adaptation that weights the different features, using an attention-like mechanism. One could propose that a neural network should be able to learn weights. However, a neural network is not inherently built to perform a weighted count. A softmax output would find the relative importance of different features, but would not reproduce the counting algorithm above. However, a sum of TANH can combine the advantages of a weighted average and counting. We show that it indeed produces a better accuracy for some real-world sets. This is similar to a hard limit or a strong regularization parameter limiting each weight in a regression to a certain range. However, in contrast with the hard limit, the optimization will not push the results to the boundaries to reproduce the $C1$ or $C2$ models.

We have then shown using a complex clinical example that these methods are appropriate even in complex situations involving hundreds of millions of potential features, and a very small number of samples.

**Limitations.** The main limitation of MILCA is ignoring all the interactions between the features. MILCA is a naive Bayesian estimator. As such any information available in the interaction between features will be lost. This delineates the applicability of MILCA to cases with many features. In contrast, much more advanced methods are required when the input dimension is low and the bags number is large.

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

# APPENDIX

