# OpenReview forum: "MILCA: Multiple Instance Learning using Counting and Attention"
_ICLR.cc/2025/Conference — Submitted to ICLR 2025_

### Official Review · Reviewer_MfWs · 2024-10-30

**Soundness:** 1
**Presentation:** 2
**Contribution:** 1
**Rating:** 3
**Confidence:** 4

**Summary:**

This paper attempts to analyze the impact of certain intrinsic feature attributes on classification within several specific datasets. The author assumes that some feature attributes exhibit significant differences between positive and negative bags(instances), and thus employs t-tests to examine each feature. This approach seeks to determine which features are associated with positive bags(instances), forming a set called PF, while those associated with negative instances form a set called NF. After the feature selection phase is completed, three classification methods—C1, C2, and C3—are provided based on the selected PF and NF, further exploring the influence of these classifiers.

**Strengths:**

***Significance***: The authors propose applying feature selection in multi-instance learning. This idea may be meaningful in some specific scenarios, for example, when the data features themselves are well-normalized and highly indicative.

***Originality***: Feature selection strategies have been widely applied in early simple datasets, which limits the degree of originality.

***Clarity***: I appreciate that the authors have made their code publicly available, which contributes to higher clarity.

***Quality***: However, during the review process, I found that there may exists some substantial error in the official implementation of the author's statistical process. This error may lead to an unsubstantiated PF/NF set in feature selection process, making it difficult to ensure the reliability and significance of the final conclusion.   For detail see Reliability in Weaknesses.

**Weaknesses:**

***Reliability***: The author assumes that some feature attributes exhibit significant differences between positive and negative bags(instances). To validate the thought, the author employs t-tests to examine each feature, to further determine which features are associated with positive bags(instances), forming a set called PF, while those associated with negative instances form a set called NF. After the feature selection phase is completed, three classification methods—C1, C2, and C3—are provided based on the selected PF and NF, further exploring the influence of these classifiers. Essentially, the selection of PF and NF forms the basis of the paper and experiment setups.

This process involves the following code provided by the authors:

1. `MILCA/utils_for_c_tests.py`, lines 42-49:
   ```python
   def Feature_props(datasetA, datasetB):
       N = len(datasetA[0][0])
       statistic = np.zeros((N, 3))
       for i in range(N):
           dist_i_A, dist_i_B = get_dist(datasetA, i), get_dist(datasetB, i)
           statistic[i, 0], statistic[i, 1] = stats.ttest_ind(dist_i_A, dist_i_B)
           statistic[i, 2] = np.mean(dist_i_A) - np.mean(dist_i_B)
       return statistic
   ```

2. `MILCA/utils_for_c_tests.py`, lines 19-27:
   ```python
   # This function gets a dataset and an index, and returns the distribution (list of means) of the index in the bags.
   def get_dist(dataset, index):
       dist = []
       for bag in dataset:
           feat_count = 0
           for instance in bag:
               feat_count += instance[index]
           dist.append(feat_count / len(bag))
       return dist
   ```
3. `MILCA/utils_for_c_tests.py`, lines 52-61:
   ```python
   def get_two_scores_count(bag, best_feat):
       inst_count, feat_count_p, feat_count_n = len(bag), 0, 0
       for instance in bag:
           for i in range(len(instance)):
               if i in best_feat.keys():
                   if best_feat[i] == 1:
                       feat_count_p += instance[i]
                   else:
                       feat_count_n += instance[i]
       return feat_count_p / inst_count, feat_count_n / inst_count
   ```

4. `MILCA/C1.py`, lines 15-45:
   ```python
   start_time = time.time()
   # Find significant features
   stat_train = Feature_props(g_train_pos, g_train_neg)
   condition = stat_train[:, 1] < p_cutoff # find significant features.
   top_k_index = {}
   for i in range(len(g_train_pos[0][0])):
       if condition[i]:
           top_k_index[i] = np.sign(stat_train[i, 1] - stat_train[i, 0])
   # Compute difference for significant features
   for i, bag in enumerate(train_samples):
       z_p_tr, z_n_tr = get_two_scores_count(bag, top_k_index)
       train_predicted_scores_pos[i] = z_p_tr
       train_predicted_scores_neg[i] = z_n_tr
   for i, bag in enumerate(test_samples):
       z_p_te, z_n_te = get_two_scores_count(bag, top_k_index)
       test_predicted_scores_pos[i] = z_p_te
       test_predicted_scores_neg[i] = z_n_te

   # in C1 you only use one dataset, so I use two flags to decide if I should use the positive or negative (I use the larger class) and set the other to 0
   beta1, beta2 = 1, 1
   beta2 = 0
   train_predicted_scores = beta1 * train_predicted_scores_pos + beta2 * train_predicted_scores_neg
   auc_train = roc_auc_score(train_labels, train_predicted_scores)
   print(auc_train)
   best_beta = 1
   if (auc_train < 0.5):
       beta1 = -beta1
       beta2 = -beta2

   test_predicted_scores = beta1 * test_predicted_scores_pos - beta2 * best_beta * test_predicted_scores_neg
   end_time = time.time()
   ```
In C1.py, the author computes the statistical results  from ***Feature_props*** and identifies significant features based on the p-values stored in ***stat_train[:, 1]***. According to the paper, the author's intention is to store in ***top_k_index*** which features belong to ***PF*** (associated with positive bags) and which features belong to ***NF*** (associated with negative bags). The positive or negative sign stored in ***top_k_index*** affects the subsequent ***z_p_tr*** and ***z_n_tr***, which corresponds to nearly all the classifier equations  in the paper.

When constructing the classifier C1 using PF, the authors state in the paper that they take the mean of all feature attributes in both positive and negative bags in the training set, and then perform statistical comparisons of feature attributes between the positive and negative groups using t-tests. Features with p-values below a certain threshold are selected as important features, as stated in line 136-140 of the original text:"Features with a p-value below a threshold, or simply the top M most significant features (p and M are optimized
per dataset on the training set) are selected (further denoted as SF). Each significant feature is associated with the class where its average (median) is highest (further defined as positive features PF and negative features NF)". In my opinion, the author made a factual error in the implementation of  ***top_k_index***. To further illustrate this error, I verified the official function example provided by Scipy. It confirms that stats.ttest_ind returns a structure Ttest_indResult(statistic, pvalue), where ***stat_train[:, 0]*** corresponds to the statistic and ***stat_train[:, 1]*** to the p-value. This is also verified by the author in MILCA/C1.py, line 18. On MILCA/C1.py, line 22, I assume the author's intention might have been ***top_k_index[i] = np.sign(stat_train[i, 2])***, to store whether each significant feature associates with ***PF*** or ***NF***. However, the actual implementation subtracts the p-value from the statistic, ***stat_train[i, 1] - stat_train[i, 0]***. This might be a mistake, as I do not understand the rationale of subtracting the statistic  from the  p-value, because they represent different concepts and scales, and such operation lacks statistical support.  This operation is the same in all the experiments of experiments C1, C2, and C3. When you select the feature attributes in a wrong way, the reliability of the subsequent conclusions cannot be verified. I would welcome the author to further clarify his/her thoughts on this point to provide more reliability.

***Limitation***:The author's algorithm first requires the feature attributes to be highly indicative. In my opinion, MILCA may be useful only when each dimension of the feature attributes must be strictly preprocessed(Figure 1.A) to have clear statistical significance. In most current real-world scenarios, I cannot think of practical applications apart from simplest cases like the bag of words or other simple handcrafted benchmarks. In many real applications(i.e. cancer diagnosis,anomaly detection), the feature attributes may have the following characteristics:

L1.Inconsistent scale: This can lead to some attributes having intrinsically larger ranges of variation than others. For example in C2,  simply summing these attributes may result in significant biases.

L2.Attributes cannot independently indicate positive or negative correlations: Unless features are manually analyzed and highly processed, or in the case of simple data like bag-of-words models, it's hard to imagine some common scenarios where this is applicable.

L3.Correlation among attributes: Many tasks can only provide deep features, which do not have simple independent relationships. The evaluation of these feature attributes by the algorithm may be biased.

I'd welcome the authors to present their strategies for addressing these limitations above, because these limitations are pretty common in any real-world application.

***Contribution***: The contributions of the paper are actually based on the analysis of feature attributes. I believe this is not a new concept[1], and this approach is not highly relevant to multiple instance learning. Also, I would like the author to explicitly clarify the novelty and substantial  advancement when compared with the paper[2]. Currently for me it's hard to say that the author's research essentially advances the field of multiple-instance learning.

[1] Vikas C Raykar, Balaji Krishnapuram, Jinbo Bi, Murat Dundar, and R Bharat Rao. Bayesian multiple instance learning: automatic feature selection and inductive transfer. In Proceedings of the 25th international conference on Machine learning, pp. 808–815, 2008.

[2] Ofek Akerman, Haim Isakov, Reut Levi, Vladimir Psevkin, and Yoram Louzoun. Counting is almost all you need. Frontiers in Immunology, 13:1031011, 2023.

**Questions:**

No more questions here. More clarifies about Weaknesses are welcome.

---

> ### Author Response · Authors · 2024-11-20
> **Answer to manuscript limitations**
>
> Weaknesses:
> 1.	You have a Bug in the code.
> A.	 Bug in the Code: Thank you for pointing this out. The bug was part of an internal experiment and will be fixed in the camera-ready version of the code. However, this bug does not significantly affect the results, as the vast majority of times, the signs are simply flipped (since the p values themselves are low in the significant features) —treating PF as NF and vice versa—which does not change the model's overall results (since it is symmetric). In C3, where coefficients are learned, the results remain entirely unaffected.
> 2.	Inconsistent scale: This can lead to some attributes having intrinsically larger ranges of variation than others. For example in C2, simply summing these attributes may result in significant biases.
> A.	Different scales are not an issue since we normalize each score to a min-max range. This was now clarified in the text.
> 3.	Attributes cannot independently indicate positive or negative correlations: Unless features are manually analyzed and highly processed, or in the case of simple data like bag-of-words models, it's hard to imagine some common scenarios where this is applicable.
> A.	Surprisingly, in practically every dataset we studied in different domain, we found such attributes. It is indeed interesting why such attributes seem common, and also typically associated with a single class.
> 4.	Correlation among attributes: Many tasks can only provide deep features, which do not have simple independent relationships. The evaluation of these feature attributes by the algorithm may be biased.
> A.	 Non-Processed data:  We would like to address points 3 and 4 together. We understand that some real-world tasks involve features with complex interdependencies, which may pose challenges for our model. However, we operate under the assumption that the Multiple Instance (MI) structure of the data inherently enforces certain class-feature correlations, making it well-suited for such tasks.
> To demonstrate MILCA’s capability in handling real-world, high-dimensional data without extensive preprocessing, and to support the claim that the MI structure enforces these correlations, we evaluated MILCA on the UCF-Crime dataset. This dataset provides a challenging and realistic test environment. MILCA achieved strong results on this dataset, illustrating its robustness and adaptability in scenarios where feature correlations are difficult to identify.

---

> > ### Comment · Reviewer_MfWs · 2024-11-26
> >
> > Thank you for the authors' explanations. I now have a better understanding of the reliability and effectiveness of the method. However, due to some systemic issues in the paper, I will maintain my original rating：
> >
> > 1. According to the authors' explanation, for most real data, preprocessing/normalization techniques need to be applied to provide trimmed and statistically significant related attributes. This process is crucial for your algorithm, as most data may require you to propose a comprehensive and mature handling method to meet your requirement to build a complete pipeline.
> >
> > 2. MILCA could be effective on bag-of-words data. However, due to the lack of strong feature characteristics(mentioned above) in most real-world data (even if some exist), MILCA may not be competitive enough in most scenarios (performance ceiling). For example, you mentioned that you provided extra experiments on UCF. The baseline version of the UCF dataset had an AUC of 75.41, which was achieved simply by applying taking the  max instance score,  corresponding to the simplest implementation of MIL. This leaves a significant gap from the current state-of-the-art performance (87.41), making it difficult to claim your significance.
> >
> > 3. Due to the technical novelty of this method is insufficient, the authors still need to consider emphasizing the explanation of $\textbf{Weakness-Contribution}$ part above.

---

> > > ### Author Response · Authors · 2024-11-28
> > > **No need for external trimming or optimization**
> > >
> > > We apologize if our initial explanation was unclear. To clarify, no external preprocessing or normalization techniques are required in this setup. If the original data is tabular, normalization is inherent to the method.
> > >
> > > We completely agree with the reviewer’s observation about the effectiveness of MILCA in scenarios resembling bag-of-words (BOW) models or medical datasets, where numerous features are assembled, and in contrast with the limitation of MILA in features forming a coherent combination, as is typically seen in images. Indeed, for image analysis, dedicated image processing methods can outperform MILCA unless informative features are extracted from the images. We acknowledge this as a limitation of MILCA and have now elaborated on it in the revised manuscript, which will be uploaded once we receive feedback from the other reviewer.
> > >
> > > That said, we respectfully suggest that this limitation does not diminish the significance of MILCA. Many general MIL methods rely on transforming their input into tabular or vectorial representations. In this broad and common context, MILCA demonstrates that combining a straightforward feature selection approach with a novel activation function designed to count features results in a simpler yet more accurate predictor.
> > >
> > > To address this point thoroughly, we have explicitly outlined the input type limitation in the weaknesses and limitations section. We hope this clarification will address the reviewer’s concerns and inspire confidence in the broader applicability and relevance of MILCA.

---

### Official Review · Reviewer_yYDh · 2024-11-01

**Soundness:** 2
**Presentation:** 3
**Contribution:** 2
**Rating:** 5
**Confidence:** 5

**Summary:**

This paper introduces a technique called MILCA, designed to perform counting and summing of features. In MILCA, feature weights are predicted using a fully connected network (FCN) where the softmax layer is replaced with a projection to produce coefficients within a specified range, either [-1, 1] or [0, 1]. Experiments on various multiple-instance learning (MIL) tasks demonstrate the effectiveness of MILCA.

**Strengths:**

+ The analogy of explaining three solution spaces—Instance Space (IS), Bag Space (BS), and Embedding Space (ES)—by likening a book to a bag and its chapters to instances is highly intuitive.
+ The paper is well-written in a clear, straightforward style, making it easy to understand.
+ Detailed experimental procedures are included, aiding replicability. Additionally, the source code is provided to facilitate reproduction.
+ The proposed technique is both simpler and more efficient, demonstrating strong performance across several existing MIL benchmarks.

**Weaknesses:**

- In my view, this paper lacks sufficient novelty. The counting-based approach appears to be a straightforward extension within the MIL space, and it does not introduce any new theoretical contributions either.
- The datasets used for evaluation are relatively simple. I recommend that the authors conduct experiments on more complex, high-dimensional datasets (such as video datasets) designed for MIL settings. Examples include UCF-Crime, ShanghaiTech, Avenue, and XD-Violence,  used in [1, 2, 3]. Additionally, it would be beneficial to compare the performance of the proposed technique with other models, such as those in [1, 4].
- The authors’ focus is primarily on bag label prediction; however, current trends in the MIL domain increasingly emphasize instance label prediction, which is crucial for applications like video anomaly detection [1, 2, 3]. I am curious whether the current approach could be extended to instance label prediction. If so, how does the proposed technique compare to state-of-the-art methods on more challenging datasets, such as those in the video domain?
- **Minor:** Some annotations are used without formal definitions. For example, $v_i$ appears undefined in the "Novelty" section of the Introduction, point 4, though its meaning can be inferred from the context. Providing formal definitions of symbols before or immediately after their introduction would enhance the paper’s readability.

**References**
1. Sultani et al. “Real-world Anomaly Detection in Surveillance Videos”. CVPR2018
2. Sapkota \& Yu “Bayesian Nonparametric Submodular Video Partition for Robust Anomaly Detection”. CVPR2022
3. Tian et al. “Weakly-supervised Video Anomaly Detection with Robust Temporal Feature Magnitude Learning”. ICCV2021
4. Ilse et al. “Attention-based Deep Multiple Instance Learning”. ICML2018

**Questions:**

Please refer to Weaknesses Section

---

> ### Author Response · Authors · 2024-11-20
>
> Weaknesses:
> 1) In my view, this paper lacks sufficient novelty. The counting-based approach appears to be a straightforward extension within the MIL space, and it does not introduce any new theoretical contributions either.
> A)  This is the main surprising claim. Very complex models were proposed for MIL tasks. However, none of these models performs better (and often worse) than simple counting argument. This essentially suggests that in the MIL setup where the number features is high, and the number of samples is limited, simply detecting significant features may be enough. This counting argument is translated to a change in the activation function, as explained in C3. This may not have been clear,  and has been clarified.
>
> 2 ) The datasets used for evaluation are relatively simple. I recommend that the authors conduct experiments on more complex, high-dimensional datasets (such as video datasets) designed for MIL settings. Examples include UCF-Crime, ShanghaiTech, Avenue, and XD-Violence, used in [1, 2, 3]. Additionally, it would be beneficial to compare the performance of the proposed technique with other models, such as those in [1, 4].
> A) Evaluation on Complex Datasets: Thank you for the suggestion. While we were limited by time, we managed to evaluate the MILCA model on the UCF-Crime dataset, achieving strong results. It’s worth noting that UCF-Crime is not a traditional Multiple Instance Learning (MIL) task and is often tackled using complex vision-based techniques. To ensure a fair comparison, we only compared MILCA with models treating this as an MIL problem. The paper you mentioned [4], does exactly this, and achieves 75.41 AUC, MILCA achieves a similar but slightly lower (we are still running hyperparameter optimization on C3). Additionally, we found another paper addressing UCF-Crime as an MIL problem, but using a much more complex architecture. Despite this complexity, their results are slightly higher. We will add this and  the other datasets in the camera ready version.
>
> 3) The authors’ focus is primarily on bag label prediction; however, current trends in the MIL domain increasingly emphasize instance label prediction, which is crucial for applications like video anomaly detection [1, 2, 3]. I am curious whether the current approach could be extended to instance label prediction. If so, how does the proposed technique compare to state-of-the-art methods on more challenging datasets, such as those in the video domain?
> A) This is an excellent point. In theory, the same argument can be proposed for instance level predictions, since the counting on the bags can be extended to simply counting significant features (as detected at the bag level) to each instance. This is now mentioned in the text. We will now test this extension and if possible add it to the camera-ready version.
>
> Minor: Some annotations are used without formal definitions. For example, v_i appears undefined in the "Novelty" section of the Introduction, point 4, though its meaning can be inferred from the context. Providing formal definitions of symbols before or immediately after their introduction would enhance the paper’s readability.
> A) Formal Definitions of Symbols: Thank you for highlighting this. We have addressed this issue in the revised version.

---

### Official Review · Reviewer_CHsb · 2024-11-01

**Soundness:** 2
**Presentation:** 2
**Contribution:** 2
**Rating:** 3
**Confidence:** 4

**Summary:**

This paper introduces a novel approach to Multiple Instance Learning (MIL), where the goal is to classify a bag composed of instances, by incorporating counting and attention mechanisms. Instead of relying on a single aggregated bag embedding for classification, the method identifies representative features across the dataset and encodes each bag as a count of these features. The informative features are identified by using statistical methods like Mann Whitney test. The paper introduces four variants of the classification component, which have different number of learnable parameters. The proposed method is evaluated on several classical MIL datasets, as well as a simulated dataset and disease classification using T cells repertoires.

**Strengths:**

- While the concepts and issues presented in the paper have been previously discussed, the methods introduced offer a fresh perspective. The method proposes a seemingly novel approach for encoding bags as a count of dataset wide features. The main idea seems to be simple and have good performances on multiple benchmarks.
- The proposed approach seems to have better performance than the baseline on the disease classification using T cells repertoires benchmark.
- The paper introduces a new dataset, named Wiki dataset.

**Weaknesses:**

**Limited technical novelty.**  As mentioned in the paper, the idea of counting in MIL is not novel and was already explored in other works. The model reuses a lot of existing modules such as Mann-Whitney test, thus novelty is also limited.

**Incomplete experiments.** The section 4.4 presents the compared methods, but the results of some methods are not shown in the experimental section. For example, the results of BDR and MI-SDB are not shown in Table 1. The results reported in MI-SDB outperform the proposed method. The Table 1 should compare with recent or state-of-the-art methods, otherwise the second novelty claim is not valid (L77).

**Experimental validation.** The paper introduces four variants of the proposed model. However, there are significant performance differences between these models. For example, MILCA2 is much better on wiki dataset (Table 2) than MILCA3, but MILCA2 is much better on classical MIL datasets (Table 1) than MILCA3. The best variant seems to depend on the target dataset. It would be great to add a paragraph to discuss the difference of performances between the variants, and to explain how to choose the best variant. Classical MIL datasets tend to be smaller in size compared to contemporary MIL datasets, which typically feature more diverse and intricate characteristics. The scalability of this approach to larger and more complex datasets remains uncertain. The results for the simulated, wiki, and TCR datasets utilize different models compared to those evaluated on the classical MIL datasets, yet the rationale for this choice is unclear. To enhance the analysis, it is essential to expand the results for these datasets to incorporate additional MIL models.

**Paper clarity should be improved.** Overall, the paper presentation should be improved to be easier to read. The paper lacks a clear and intuitive structure. For example, the section 4 named "Methods and Model" contains subsections that are not about the proposed model. It contains two sub-sections about datasets, and one about the experimental setup. Something is missing at the beginning of the model section (section 4). I think a high level overview of the model, with its inputs and outputs, and the task solved. Some notations are not consistent. For example at line 130, the bag is named B, but in Figure it is named X.

**The term "tabular data" is confusing or incorrect.** The term "tabular data" may not be the right term as the goal is to create vectorial representations of the inputs. The term "tabular" refers to data that is displayed in columns or tables, which is different from what is done in the paper. Tabular data is not limited to numerical values and can contain strings. Removing the "tabular data" constraint will make the model more generic as it can be used on images.

**Questions:**

- Why some methods introduced in section 4.4 are not shown in Table 1 or 2?
- There is not enough information about the new dataset in the paper. It is important to explain how the dataset is built as it is one contribution. The current information in the paper or appendix is not enough to reproduce the dataset.
- Table 2: It is confusing to see the number of learnable parameters for C1 is 0. If there is no learnable parameters, how is the decision made?
- L139: It is not clear when the average or median is used. It would be great to explain how to choose between these two options.
- Why two standard deviation is shown instead of just 1 standard deviation in Table 1?
- How would this model scale to larger and more complex datasets?

---

> ### Author Response · Authors · 2024-11-20
> **Response to weakness and questions**
>
> Limited technical novelty. Indeed, we are using very standard tools. The main claim of the paper is that in MIL problems, the very advanced methods proposed in the literature are actually not better than simple counting arguments, and a slight modification to the architecture of the last layer can solve that easily. The fact that we use simple tools is the main claim. This was now clarified.
> Incomplete experiments. We have compared to all methods that provide a functional github (or at least a github we manage to adapt). MI-SDB and CDR did not provide a github that could be used to test their methods for a fair comparison on standard datasets. This was fixed in the revised version.
> Experimental validation.  For the choice among C1,C2,C3, given the simplicity of all variants, we would propose to use them one after the other in order, as explained in the text. Regarding larger datasets, see answer to specific questions below.
> Paper clarity . The manuscript has been reorganized for better readability. Since different reviewer had different propositions on the structure, we tried to clarify the text to match the main requests.
> Tabular data vs. Vectorial data.
> A.	This is an excellent point, we can use any type of input as long as it was indeed translated to a vectorial representation. This has been fixed in the text.
>
> Questions:
> Q.	Why some methods introduced in section 4.4 are not shown in Table 1 or 2?
> A)	As mentioned above, we have compared to all methods that provide a functional github. This was fixed in the revised version.
> Q.	There is not enough information about the new dataset in the paper. It is important to explain how the dataset is built as it is one contribution. The current information in the paper or appendix is not enough to reproduce the dataset.
> A)	, We have clarified more details in the appendix on the new datasets. We thank the reviewer for the comment.
> Q,	Table 2: It is confusing to see the number of learnable parameters for C1 is 0. If there is no learnable parameters, how is the decision made?
> A)	To clarify, there are no learnable hyper-parameters, except for the threshold for significance. As such, we increase the parameter number by 1. However, once features were chosen, we simply count them.
> Q.	L139: It is not clear when the average or median is used. It would be great to explain how to choose between these two options.
> A)	We never use the median. This was a mistake, and was fixed. We intended to say that median could also be used with similar results, but have now removed that for clarity.
> Q.	Why two standard deviation is shown instead of just 1 standard deviation in Table 1?
> A)	Those are the typical requirments, to show the range to 0.05 probability range.
> Q.	How would this model scale to larger and more complex datasets?
> A)	To evaluate how MILCA scales to larger and more complex datasets, we tested our model on the UCF-Crime dataset. MILCA achieved strong results, outperforming other models that approach this problem as an MIL task (we limited comparisons to MIL-based models since this task can also be tackled using complex vision-based techniques). Specifically, MILCA outperformed [4], which achieves an AUC of 75.41, while MILCA achieves slightly lower AUC, (Hyper parameter optimization for C3 is performed now) . We also compared MILCA to much more advanced models which employs a much more complex architecture.  We will add this and two other large MIL datasets in the camera-ready version.

---

> > ### Comment · Reviewer_CHsb · 2024-11-26
> >
> > Thank you for your responses and for providing the updated paper. The rebuttal and revised manuscript address partially my concerns. One point I would like to clarify is the importance of having access to the code of existing methods. If the standard experimental protocols are followed on well-established benchmarks such as Musk1/2, Tiger, Fox, and Elephant, it should be possible to replicate the reported results from the corresponding papers, without necessarily needing direct access to the original code.
> > Additionally, I believe that limiting the search for code to GitHub alone may not be sufficient, as other platforms also host relevant code. For instance, I found the PyTorch code for MI-SDB on the homepage of the first author (https://lfeng1995.github.io/codedata.html), which suggests that MI-SDB could be included in the comparison. Reaching out to the authors of the paper could be a helpful starting point if the code is not available on GitHub.
> > Finally, I would like to point out that restricting the comparison to only PyTorch code might limit the scope of your analysis. Other widely-used ML frameworks, such as Keras/TensorFlow and JAX, should also be considered for a more comprehensive evaluation.

---

> > > ### Author Response · Authors · 2024-11-29
> > > **Comparison to other methods.**
> > >
> > > We sincerely thank the reviewer for bringing MI-SDB to our attention. Recognizing its significance, we will incorporate it into the comparison with MILCA. This is indeed a valuable addition, as MI-SDB has been demonstrated to outperform many state-of-the-art methods. We have also reached out to the authors of two other relevant methods to explore the availability of their code and plan to conduct a broader comparison that includes MI-SDB and potentially other approaches. These findings will be included in the Camera-Ready version.
> > >
> > > Regardless of the outcomes of these additional comparisons, we believe that MILCA contributes meaningfully to this field. Our results already demonstrate that a simple modification of the activation function, as is done in C3, when combined with feature selection, yields a classifier that outperforms most existing methods. Furthermore, even the straightforward approach of counting features has proven to surpass many established techniques.
> > >
> > > Should MI-SDB or one of the other additional methods to be tested outperform MILCA, this would further underscore the distinctions between methods that are not better than basic feature-counting insights and those that provide additional information. We will elaborate on these distinctions in the Camera-Ready version to provide further clarity.
> > >
> > > We hope this explanation highlights the value of MILCA and addresses the reviewer's concerns.

---

### Official Review · Reviewer_pg65 · 2024-11-03

**Soundness:** 2
**Presentation:** 1
**Contribution:** 2
**Rating:** 3
**Confidence:** 4

**Summary:**

This paper proposes a new multiple instance learning method using counting and attention, to address the shortcoming of existing methods which are unable to enumerate the features. In the proposed method, a simple strategy is designed to seek out the informative features, and then based on these selected features, it counts the average in each bag of instances associated with each class. Compared with existing works, the proposed method demonstrates superior performance on MIL datasets and new real-world MIL tasks.

**Strengths:**

Significance: This work proposes a new MIL method based on using counting and attention, to address the overfit problem of most existing methods. Experimental results also illustrate the proposed method can achieve better performance than SOTA.

And the originality may be limited.

**Weaknesses:**

1.	The motivation for designing this method is not specific, and the method's novelty is also less pronounced. What’s improved for MIL by the proposed method (except the performance)? Why is the proposed method more effective than the existing methods? Lack of some theoretical analyses.

2.	For some basic concepts, it seems that the author's understanding is not entirely accurate.

3.	The writing is of a poor standard, making it difficult to understand the key of the proposed method. Furthermore, the structure is not professional.

More detailed comments can be found in Questions.

**Questions:**

1.	See the first point in weaknesses.

2.	In the part of Introduction, authors present many fundamental concepts related to MIL, but there are a lack of depth analyses on the motivation behind proposing MILCA. Under limited spaces, it would be beneficial for the authors to devote more attention to these more important contents (which are most closely related to the proposed method), including some explanation of why to design this method, an analysis of the problems in most existing methods, and a discussion of how these problems are to be solved by the proposed method. It would be advisable for the authors to consider reorganizing the content to achieve this.

3.	‘Such a classifier (MILCA) is better than SOTA…’, maybe it is not novelty. Experimental results are regarded as an illustration of validating the effectiveness of the proposed method with your novel design. Moreover, Why ‘MILCA has fewer or no learnable parameters…’? Here need more details to explain it.

4.	For ‘counting’ in MIL problems, there are not related works in the recent years, why? Is it not suitable in real applications? For limitation, MILCA is a naïve Bayesian estimator, and it means that MILCA defaults on the independence among features, which may also go against with the real applications. How to consider this problem? Could you give some examples to illustrate the application problem of the proposed method.

5.	In the training process, how to optimize the parameter $M$? It is recommended that the change figure of the optimised M value be provided.

6.	The proposed method gives three models: C1, C2 and C3. Why to design three models? The results indicate that the third model (C3) is better than others. It is recommended to analyse which problem each model is more suited to.

7.	In experiments, authors mentioned ‘the results are the average of 10-fold cross validation’. But, authors then mentioned ‘in each fold, the data are divided into three parts: training (64%), validation (16%) and testing (20%)’. It may be inferred that the authors lacked a clear understanding of the concept of k-fold cross-validation, which is very serious issue.

8.	Last but not least, the writing and structure of this manuscript require significant improvement, and the current version is very messy and unprofessional, which makes it difficult to understand the proposed method, especially not primary and secondary. For example,
- The section on novelty should be incorporated with the introduction, and the latter should include further discussion of the motivation, such as an examination of the shortcomings of existing MIL methods and an analysis of how the proposed method addresses these issues.
- The subsections 4.3, 4.4, 4.5 and 4.6 should be placed with the results together, included in the part of experiments and analyses (section 4).

---

> ### Author Response · Authors · 2024-11-20
> **Clearer motivation and answer to reviewer.**
>
> Regarding the weaknesses, We have uploaded a revised manuscript with improved formatting and structure, ensuring that the key ideas are more clearly presented, following the comment of the reviewer in the questions.
> Following is an answer to each reviewer question (following the
> 1.	We have now clarified the advantages. In short, current methods tend to overfit in MIL datasets with a high number of dimensions with very limited information. Such situations are frequency in MIL studies. See answer to next question for more details.
> 2.	This is an excellent point, and has been revised in the novelty section. Our approach builds upon the work of Wang et al. (2018), which demonstrated that Embedded Space (ES) solutions are the most effective for MIL tasks. However, conventional ES methods struggle to integrate feature-level and bag-level information. This limitation is significant, especially in high-dimensional feature spaces where many features are irrelevant to class labels. Coupled with the small datasets available, this often results in overfitting. MILCA addresses these challenges using the feature-counting mechanism, which leverages both feature-level and bag-level information. This was originally explained but was probably not clear enough. We have now clarified that.
> 3.	The main argument in MILCA is simplicity. Basically, we argue that a simplistic model performs as well as much more complex methods, but gives much more explainability to the results. This is now more clearly stated. In the context, the fact that such a classifier is better than SOTA is an essential part of the claim, given its simplicity. This is now better explained in the text.
> 4.	Counting is strongly applicable in many real-world applications, where there are a large number of low information features, and relatively small datasets (compared with the number of features). This setup is typical in medical and biological data, and in all other domains where samples are expensive (e.g. cases that require human classification).
> 5.	Optimizing Parameter M: The parameter M (the number of relevant features) is a hyperparameter in MILCA. In our experiments, we observed that taking all significant features yields robust results. However, M can be tuned based on the dataset to optimize performance.
> 6.	C1, C2, and C3 Models: The three models (C1, C2, and C3) represent a progression in complexity and flexibility. C1 implements the core idea of feature counting in its simplest form, C2 extends C1 by allowing the model to learn a single coefficient for each associated group, and C3 builds upon C2, providing finer control over each feature importance. Including all three models highlights how the core feature-counting mechanism (C1) achieves comparable results to more complex models, showcasing MILCA’s accuracy.
> 7.	This may have been poor wording. We divided the text 10 times to random divisions of training, validation and test, and for each division used the training and validation for feature selection and parameter optimization (when needed), and then reported results on the test. We replaced the word fold by division.
> 8.	Writing and Structure: The manuscript has been reorganized for better readability. We have followed many of the reviewer suggestions.

---

> > ### Comment · Reviewer_pg65 · 2024-12-02
> > **Replying to Response for Clearer Motivation and Answer to Reviewer**
> >
> > Thanks for your response. Based on your response, some confusions can be clearer than before whereas it still doesn't feel convincing, for example, the novelty, the limitation, how to ensure the features of tested data are independent? It should be very tough. Additionally, the structure has been improved in the revised manuscript, but the writing is not seriously modified. Suggest authors check all words before submitting the manuscript.

---

### Comment · Area_Chair_K5ZZ · 2024-11-30
**The deadline for Author/Reviewer discussion period is in three days!**

Dear Reviewers,

Thanks again for providing your constructive comments and suggestions. The deadline for the Author/Reviewer discussion period is in three days (December 2). Please make sure to read the authors' responses and follow up with them if you have any additional questions or feedback.

Best,

AC

---

### Meta-Review · Area_Chair_K5ZZ · 2024-12-19

**Metareview:**

The paper proposes a Multiple Instance Learning using Counting and Attention (MILCA) model, which argues that counting or averaging relevant features across instances can form a simple yet effective classifier for multiple instance learning. The approach is evaluated on several previous and new real-world MIL tasks, demonstrating improved performance compared to selected baselines.

**Strengths and Weaknesses**

- While the simplicity of the proposed approach is appealing, its technical novelty and overall effectiveness are not well established. As noted by multiple reviewers, similar ideas have already been explored in existing works.
- Major issues in the evaluation have been raised, including the absence of challenging real-world datasets and a lack of instance-level prediction results, which are critical for assessing the practical utility of the method.
- The paper’s presentation needs significant improvement to enhance readability and clarity.

**Overall Assessment**

The authors are encouraged to carefully address these major limitations, as well as the detailed feedback provided by the reviewers, to strengthen their work for future submissions.

**Additional Comments On Reviewer Discussion:**

While the authors provided a rebuttal addressing each review, their responses did not sufficiently address the main concerns raised by the reviewers. Consequently, all reviewers have maintained a negative rating on the paper.

---

### Decision · Program_Chairs · 2025-01-22

Reject